# Particle Size and Potential Toxic Element Speciation in Municipal Solid Waste Incineration (MSWI) Bottom Ash

**Luciana Mantovani** [1,*] , **Mario Tribaudino** [1] , **Chiara De Matteis** [1] **and Valerio Funari** [2]

1   Dipartimento di Scienze Chimiche, della Vita e della Sostenibilità Ambientale, Università di Parma, 43124 Parma, Italy; mario.tribaudino@unipr.it (M.T.); chiara.dematteis@unipr.it (C.D.M.)
2   Consiglio Nazionale delle Ricerche, Istituto di Scienze Marine, ISMAR-CNR Bologna Research Area, 40129 Bologna, Italy; valerio.funari@bo.ismar.cnr.it
*   Correspondence: luciana.mantovani@unipr.it

**Abstract:** The speciation of potentially toxic elements (PTE) in bottom ashes from municipal solid waste incineration (MSWI) and their relationship with grain size is investigated. The proposed enrichment of several potential toxic elements in lower sized grains is discussed, comparing the literature and new data on Parma's waste incinerator. Results from X-ray fluorescence spectrometry (XRF), SEM-EDS, and XRD analyses on different grain size show (1) a positive Si-trend, correlated with grain size and few lithophile elements, such as Zr and Rb. In Parma, Al, K, Mg, and Fe also correlate with Si for the portion below 2 mm; (2) a Ca-trend, with a strong negative correlation with Si and a positive correlation with loss on ignition (LOI), S, Cl, Ti, Zn, Pb, and Sn. Mineralogical composition shows a little change in grain size, as in previous investigations, but with substantial differences in amorphous content. SEM-EDS analysis shows that the amorphous portion is highly heterogeneous, with portions coming from melting during incineration, residual glass, and unburnt loss on ignition (LOI). The above results show that PTE elements are either present as metals (such as Cu and Ni, or Zn, Pb and Sn) in carbonate, sulfate, and amorphous residual LOI portions.

**Keywords:** bottom ash; mineralogical characterization; potential toxic elements; grain size distribution; XRD; SEM-EDS; XRF

## 1. Introduction

The European Union has issued several directives on the management of waste that involve the prevention of waste formation, the reduction of the new dangerous material, the reuse and recycle and, the use of waste to produce energy. These were implemented by the national laws, e.g., in Italy by the d.lgs. 152/2006 [1].

The most recent communication from the European Commission is COM (2014)394 "Towards a circular economy: a zero-waste programme for Europe". It proposes the recycling of 70% urban and 80% packaging wastes by 2030 and forbids landfill disposal of any recyclable waste from 2025 [2]. However, several waste categories cannot be recycled (i.e., hazardous waste with European Codes EWC signed with *). In order to recover some material and energy from the combustion process, waste-to-energy incineration plants are increasing in number. In the European Union, facilities burn over 131 kilos of waste per capita per year, but a very close fraction (117 kg per capita) ends in landfills [3]. A major increase in waste burning is foreseen since, within a few years, all non-recycled waste must be reduced as much as possible to avoid landfilling.

Incinerator plants produce combustion residues in the form of bottom ashes (BA), about 20% of the total waste mass, and in the form of fly ashes, about 4% [4]. In both cases, there is a growing need for the reuse of waste incinerator products. The current objective of reusing waste prompts the scientific community to analyze the chemical (composition, mineralogical phases, and heavy metal content) and physical (particle size distribution, bulk density, permeability, and porosity) properties of these ashes.

BA are very heterogeneous materials. They are composed of silicates, mainly in the form of amorphous glass; oxides; metallic part; and up to 5% unburned organics, but carbonates, sulfates, and chlorides are also present [5–9]. The actual chemical and mineralogical composition of BA varies with respect to plant type, country, collection area and policy, season, etc.

A detailed analysis of the mineralogical composition has been conducted by different authors [5,10–13], but only in a few cases are the most recent plants referred to [14–16]. In fact, most investigations on BA focus on reuse and potential environmental hazards [17–20].

Little or no consideration is generally paid to which phases host critical elements. Namely, it is not clear where metals and heavy metals in higher concentration, such as Cu, Zn, and Pb, are found, whether in a small reactive host, like a crystalline oxide, or in a more reactive one, such as glass or carbonate. This could become an important issue for reuse: BA are industrial non-hazardous waste (2008/98/CE), but in the presence of reactive mineral phases, with heavy metals embedded in their crystalline or amorphous structure, the ash would not be suitable for reuse. A focal point debated after the European Council Regulation 2017/997 concerns the heavy metals glass and mineral-bearing phases, their identification and characterization, and their weathering behavior [21].

It is established that the smaller portion of the bottom ashes is the more polluted [14,22]. This was systematically verified in a recent investigation that sorted the BA in a range of different grain sizes. BA sorting produces new waste with the further disadvantage that smaller sized grains provide a larger reaction surface and, possibly, higher leaching of heavy metals already present at significant concentrations [14,15,23–25]. Recovery of the smaller fraction of the BA was proposed in regard to pyroxene glass ceramics by heating at a high temperature [26]. However, characterization through different sizes of the BA is required to plan any process of recovery.

Previous investigations focused on the characterization of the mineral phases [11,14] or the analysis of the portions sieved from bottom ashes [15,23]. A systematic investigation regarding mineralogy, composition, and grain size, to constrain the host for the potentially toxic elements (PTE) is still lacking. It is not clear whether the PTE are always concentrated in small grain sizes or whether their presence primarily depends on the waste input or the type of plant [27].

This study compares the average mineralogical and chemical composition for each fraction divided by grain size, using results from the literature and new data from the waste-to-energy plant (WtE) located in Parma. X-ray diffraction, chemical analysis of major and minor elements, and optical and electron microscopy, together with microprobe analysis observations, are used to clarify the trends followed by PTE in terms of grain size and BA mineralogy. The aim of this study is to determine how grain size affects the distribution of the different elements and any possible reuse and recycling implication.

## 2. Materials and Methods

### 2.1. Bottom Ash Sampling and Sieving

Municipal solid waste incineration (MSWI) bottom ash was collected from the WtE plant of Parma (northern Italy), which is located in the first suburbs of the city in the PAIP (Polo Ambientale Integrato per la gestione dei rifiuti di Parma—Environmental Integrated Area for waste management). Designed and built by Iren Group, the plant has been operating since 2014, covering about 150,000 tons/year of material in 2019. The main feedstock waste includes a dry fraction selected from undifferentiated municipal solid wastes (70,000 t/year); special unrecycled wastes (18,000 t/year); discharges from waste recovery and disposal (15,300 t/year); sanitary wastes (3500 t/year); cemetery wastes (200 t/year); industrial, handicraft, and commercial processes wastes (3000 t/year); and dried sewage sludge (20,000 t/year). The plant produces about 32,000 t/year of BA, as well as slag and fly ash (about 20% of the ashes).

Five samples, each weighing about 600 g, were taken, blending a larger amount of material from different points of the bottom ash pile located inside the plant on five different

days (1–5 December 2018) to consider the variability of the material. The individual samples were mixed; the total sample of about 3 kg of fresh BA was dried in the oven at 50° C for 24 h and sieved in order to assess the grain size distribution. The openings standards used were chosen according to European standards for aggregate EN 933–2 [28], and the grain size used for further investigation was divided into nine classes (<0.063, 0.2, 0.3, 0.5, 1, 2, 4, 8, >16). The cumulative grain size distribution is shown in Figure 1.

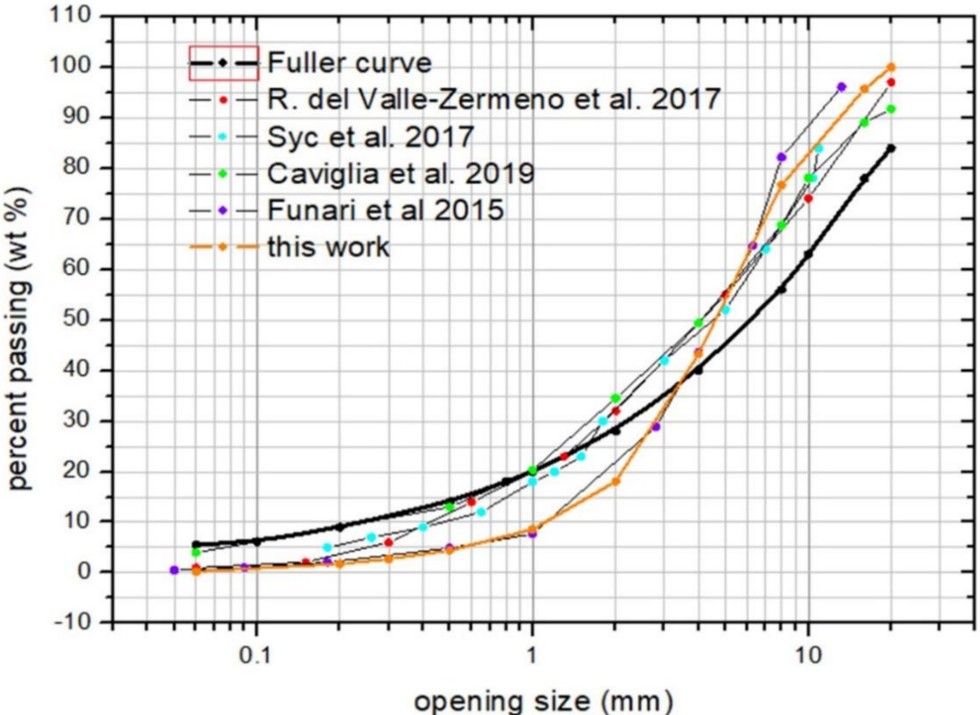

**Figure 1.** Cumulative particle size distribution and comparison with the Fuller curve and literature data [8,15,29–31].

### 2.2. X-ray Fluorescence Spectrometry (XRF)

The bulk composition was measured for each granulometric class by means of X-ray fluorescence spectrometry (XRF). About three grams of the dried and milled material was used. Before analysis, each sample was first pressed in a boric acid binder to obtain a thin-layer pressed powder pellet (37 mm in diameter). A sequential wavelength dispersive X-ray fluorescence (XRF) spectrometer (Axios-Panalytical), equipped with a 4 kW Rh tube and SuperQ 3.0 software, was used. Total loss on ignition (LOI) was gravimetrically estimated after overnight heating at 950° C. The analytical results in Table 1 are provided with measured vs. certified values of the certified reference material (CRM) BCR-CRM176R. The estimated precision for elemental determinations is better than 5% for all elements, except those occurring at concentrations lower than 10 mg/kg, for which the precision is comparably worse (10–15%).

**Table 1.** X-ray fluorescence spectrometry (XRF) analysis of major, minor, and trace elements for the bottom ashes (BA) samples of Parma waste-to-energy (WtE) plant divided by grain size. Elements are in mg/kg, but major elements are expressed as g/100 g of their oxides and loss on ignition (LOI) values in %. The measured vs. certified concentrations of the certified reference material (CRM), BCR-CRM176R, is provided with the same unit of measure (n.a. is not available).

| Major Elements (g/100 g) | Grain Size (mm) | | | | | | | | | | CRM | |
|---|---|---|---|---|---|---|---|---|---|---|---|---|
| | >16 | 8–16 | 4–8 | 2–4 | 1–2 | 0.5–1 | 0.3–0.5 | 0.2–0.3 | 0.063–0.2 | <0.063 | m.v | c.v |
| $SiO_2$ | 46.23 | 45.82 | 38.95 | 37.75 | 35.91 | 31.72 | 29.20 | 25.71 | 24.85 | 23.71 | n.a | n.a |
| CaO | 21.60 | 21.78 | 24.79 | 24.66 | 24.63 | 26.67 | 28.41 | 29.67 | 30.12 | 30.05 | n.a | n.a |
| $Al_2O_3$ | 8.59 | 8.26 | 8.92 | 10.39 | 10.05 | 9.63 | 8.96 | 8.27 | 8.46 | 8.27 | n.a | n.a |
| MgO | 3.71 | 3.95 | 3.64 | 4.33 | 4.42 | 3.91 | 3.59 | 3.26 | 3.15 | 3.23 | n.a | n.a |
| $Fe_2O_3$ | 3.04 | 3.01 | 3.60 | 3.63 | 4.05 | 3.74 | 3.48 | 2.64 | 2.49 | 2.21 | 1.6 | 1.3 |
| $Na_2O$ | 3.92 | 3.69 | 2.78 | 2.49 | 2.36 | 2.20 | 2.18 | 2.14 | 2.26 | 2.20 | 2.6 | 3.5 |
| $P_2O_5$ | 1.52 | 1.30 | 1.59 | 2.22 | 2.11 | 2.10 | 2.19 | 2.00 | 1.91 | 1.82 | n.a | n.a |
| $K_2O$ | 1.21 | 1.29 | 1.26 | 1.36 | 1.45 | 1.35 | 1.29 | 1.23 | 1.19 | 1.11 | n.a | n.a |
| $TiO_2$ | 0.69 | 0.64 | 0.78 | 0.86 | 0.90 | 0.97 | 0.92 | 0.90 | 0.90 | 0.90 | n.a | n.a |
| MnO | 0.09 | 0.08 | 0.11 | 0.09 | 0.10 | 0.10 | 0.10 | 0.09 | 0.09 | 0.10 | 0.08 | 0.08 |
| LOI | 9.40 | 10.19 | 13.60 | 12.22 | 14.02 | 17.62 | 19.69 | 24.08 | 24.58 | 26.40 | n.a | n.a |
| **Minor and trace elements (mg/Kg)** | | | | | | | | | | | | |
| S | 8070 | 7780 | 9600 | 9500 | 10,900 | 12,840 | 14,780 | 16,910 | 17,740 | 16,420 | n.a | n.a |
| Cl | 5430 | 5430 | 7420 | 8050 | 8236 | 9780 | 10,120 | 10,530 | 10,960 | 11,600 | n.a | n.a |
| Cu | 1261 | 1413 | 1104 | 1669 | 1335 | 1640 | 1885 | 1664 | 1637 | 2041 | 980 | 1050 |
| Zn | 1380 | 4400 | 1660 | 3630 | 3830 | 4270 | 5750 | 5940 | 6830 | 8740 | 15,714 | 16,800 |
| Ba | 955 | 907 | 1098 | 1684 | 1461 | 1681 | 1618 | 1631 | 1529 | 1879 | 4141 | 4650 |
| Pb | 1354 | 409 | 312 | 545 | 676 | 800 | 802 | 913 | 981 | 1172 | 5216 | 5000 |
| Cr | 880 | 629 | 573 | 434 | 651 | 644 | 697 | 592 | 621 | 627 | 794 | 810 |
| Sr | 429 | 378 | 436 | 681 | 499 | 485 | 517 | 571 | 568 | 572 | n.a | n.a |
| Zr | 256 | 212 | 186 | 207 | 196 | 173 | 169 | 155 | 156 | 143 | n.a | n.a |
| Ni | 119 | 183 | 140 | 135 | 174 | 144 | 179 | 132 | 134 | 160 | 123 | 117 |
| Co | 27 | 27 | 32 | 25 | 62 | 51 | 57 | 44 | 44 | 43 | 29 | 26.7 |
| V | 61 | 69 | 73 | 84 | 84 | 90 | 79 | 78 | 78 | 79 | 113 | 35 |
| As | 54 | 26 | 41 | 36 | 47 | 52 | 56 | 59 | 59 | 74 | 50 | 54 |
| Ce | 37 | 44 | 41 | 46 | 47 | 48 | 36 | 29 | 38 | 51 | 41 | 47.7 |
| Sn | 19 | 17 | 19 | 44 | 36 | 53 | 43 | 49 | 58 | 91 | n.a | n.a |
| Rb | 32 | 34 | 31 | 30 | 32 | 30 | 27 | 27 | 25 | 24 | 85 | 102 |
| La | 18 | 35 | 13 | 8 | 25 | 30 | 21 | 6 | 14 | 16 | 28 | 30.2 |
| Y | 14 | 16 | 16 | 17 | 19 | 15 | 14 | 12 | 12 | 11 | n.a | n.a |
| Nd | 18 | 19 | 10 | 10 | 22 | 18 | 20 | 2 | 19 | 5 | n.a | n.a |
| Mo | 15 | 10 | 15 | 11 | 17 | 14 | 13 | 15 | 14 | 14 | n.a | n.a |
| Ga | 14 | 13 | 12 | 13 | 13 | 13 | 14 | 14 | 13 | 14 | n.a | n.a |
| Nb | 11 | 10 | 11 | 12 | 12 | 11 | 10 | 10 | 11 | 10 | n.a | n.a |
| Sc | <3 | 6 | 10 | 15 | 10 | 6 | 13 | 14 | 18 | 14 | 3.4 | 2.91 |
| Th | 13 | 7 | 6 | 5 | 7 | 7 | 7 | 8 | 9 | 9 | 6 | 5.28 |
| Hf | <3 | 7 | 6 | 4 | 3 | <3 | <3 | <3 | <3 | <3 | 7 | 4.85 |
| U | <3 | 3 | <3 | <3 | <3 | <3 | <3 | <3 | <3 | <3 | n.a | n.a |

*2.3. X-ray Powder Diffraction (XRPD) Phase Analysis*

X-Ray Diffraction (XRD) was performed on each of the sieved portion, and on a few mm-sized grains chosen in terms of optical appearance. A Bruker D2 Phaser powder diffractometer with the following specifications was used: Cu Kα (λ = 1.54178 Å) radiation, 30 kV and 10 mA, Ni filtered, 2θ between 5 and 70°, steps of 0.02°, and a sampling time of 1, s. The diffractometer acts with θ-θ focalizing geometry and takes advantage of a solid-state detector. A sample rotation of 30 rpm was applied to minimize crystal preferential orientation effects. The diffraction patterns were identified using the Bruker software EVA and the Crystallography Open Database (COD). The complexity given by the number of mineralogical phases hindered quantification through Rietveld refinement of all of the phases present. The major crystalline phases, found in all samples, were quantified using the GSAS 2 software package [32]. The amorphous content was estimated by Rietveld analysis [33,34] using high=purity $Al_2O_3$ corundum (10 wt%) as an internal standard. The refined phases and their abundance are listed in Table 2.

**Table 2.** Rietveld refinement quantitative phase analysis (wt%) on different grain size. The estimated error is $\pm 1$.

| Phases | Grain Size (mm) | | | | | | | | | |
|---|---|---|---|---|---|---|---|---|---|---|
| | <0.063 | 0.063–0.2 | 0.2–0.3 | 0.3–0.5 | 0.5–1 | 1–2 | 2–4 | 4–8 | 8–16 | >16 |
| Calcite $CaCO_3$ | 10 | 11 | 10 | 9 | 8 | 8 | 5 | 7 | 5 | 7 |
| Quartz $SiO_2$ | 3 | 4 | 2 | 6 | 3 | 5 | 6 | 6 | 4 | 5 |
| Strätlingite $Ca_2Al_2SiO_7 \cdot 8H_2O$ | 2 | 2 | 2 | 2 | 2 | 1 | 1 | 1 | 1 | 1 |
| Ettringite $Ca_6Al_2(SO_4)_3(OH)_{12} \cdot 26H_2O$ | 4 | 6 | 5 | 3 | 3 | 2 | 1 | 1 | 3 | 3 |
| Hydrocalumite $Ca_4Al_2(OH)_{12}(Cl,CO_3,OH)_2 \cdot 4H_2O$ | 3 | 4 | 4 | 4 | 3 | 3 | 3 | 3 | 3 | 2 |
| Anorthite $CaAl_2Si_2O_8$ | 2 | 2 | 2 | 7 | 2 | 3 | 3 | 2 | 1 | 2 |
| Vaterite $CaCO_3$ | 3 | 3 | 4 | 4 | 2 | 3 | 1 | 2 | 2 | 2 |
| Gehlenite $Ca_2Al(AlSiO_7)$ | 2 | 3 | 3 | 2 | 2 | 2 | 2 | 2 | 2 | 2 |
| Hematite $Fe_2O_3$ | 1 | 1 | 1 | 1 | 3 | 1 | <1 | <1 | <1 | <1 |
| Magnetite $Fe_3O_4$ | <1 | <1 | <1 | <1 | <1 | <1 | <1 | <1 | <1 | <1 |
| Crystalline | 30 | 36 | 33 | 36 | 28 | 27 | 23 | 24 | 20 | 25 |
| Amorphous | 70 | 64 | 67 | 64 | 72 | 73 | 77 | 76 | 80 | 75 |

*2.4. Optical and SEM-EDS Microscopy*

Twelve large fragments (>8 mm) were embedded in epoxy resin and cut longitudinally to obtain thin and polished sections for the analysis. The embedded grains were selected to avoid unburnt and refractory materials, which were instead part of the powder analyzed by XRD and XRF. The sections were first observed by a polarized stereomicroscope (BX51, OLYMPUS) and then with a scanning electron microscope coupled with an energy dispersive system (SEM-EDS) JSM IT300LV Jeol 6400 equipped with an Oxford EDS microprobe. Microprobe analysis was performed with the following operating conditions: 20 or 25 kV and 1.2 mA current, ~1 μm beam diameter, and a counting time of 75 s. Analyses at 25 kV were performed in order to enhance the contribution of the higher energy peaks in metals and heavy metals.

*2.5. Statistical Analysis*

A descriptive statistic was applied to selected chemical and mineralogical results based on grain size as a discriminant variable, and it was used for comparative analysis with two relevant works recently carried out by Caviglia et al. (2019) and Loginova et al. (2019) [15,23]. The correlation was investigated by calculating the linear regression ($r$, Pearson) and Spearman's rank $p$ order correlation coefficient.

Statistical analysis of data distributions was performed using SPSS software V.27 [35] and is reported in Table S2.

**3. Results**

*3.1. Grain Size Distribution*

In Figure 1, the granulometric curve of the samples is shown and compared with that of other incinerators with similar technology and conditions, as reported by Caviglia et al. (2019), Funari et al., (2015), Šyc et al., (2018), and del Valle-Zermeño et al. (2017) [8,15,29,30]. The Fuller curve, which represents the optimal aggregate distribution curve in terms of density and strength, is also plotted [31]. Some of the difference may come from the sampling procedure. The curves vary depending on the portion of the sampled ash pile, overestimating the larger size in the lower portion [29], but for all the granulometric plots, the fraction lower than 4 mm accounts for about 55 wt%.

In particular, the cumulative particle size distribution of Parma shows that most of the BA (about 60%) lies in the range of 2–9 mm (range of coarse sand and gravel), while 20% of the total weight has a grain size < 2 mm, and another 20% is the fraction > 9 mm. A difference exists in the portion below 1 mm, which is just 6 wt% in Parma and Funari et al. 2015, whereas it is up to 20 wt% in other studies [30]. Comparing the shape of the Fuller

curve to our samples, we have a lower percentage—about 5–10 wt%—of fine material (below the 3 mm).

### 3.2. Chemical Analysis of MSWI Bottom Ash

The bulk chemical composition with grain size is reported in Table 1. Pearson's *r*-value and the Spearman rank *p* to test correlation between elements and with grain size are reported in Table S1.

In the Parma bottom ashes, Si is enriched in the larger grains and Ca in the smaller ones. There are several elements that are positively correlated with Si or with Ca. The positive trend with Si is followed by Rb and Zr, and, among major elements, by Al, Fe, K, and Mg, when only the portions smaller than 2 mm are considered. In this case, the correlation is strong ($R^2 > 0.9$). Several other elements are positively correlated with Ca, and negatively with grain size: they are S, Cl, Zn, Cu, Ba, Pb, As, Sn, Ti, and Sr. Few elements, Cr, Ni, V, and Ce, do not seem to be related to Si or Ca.

Thus, we find a Si-trend, which is followed by lithophyle elements, i.e., those having an affinity for Si, incorporated in silicates, and a Ca-trend, for elements with an affinity for carbonates and sulfates (Ca, Ba, Zn, and Pb), sulfides (Sn and As), oxides (Ti), or possibly to the unburnt organics (LOI, Cl, and S).

Si- and Ca-trends are also present in other papers where the composition was determined with grain size. The differences with these observations are that Loginova et al. [23] showed a strong correlation with Ca and S for Ni and Cr, and Caviglia et al. [15] showed that Ni, but not Cu, follows the Ca-trend (Table S1). Moreover, the positive correlation with Si for Fe, Mg, K, and Al in the smaller grains is not confirmed; in Loginova et al. [23], Fe has a negative correlation with Si. Within the above correlations, there is a strong difference in the actual compositional values (Figure 2). Si vs. Ca changed with a similar slope in this work and in that of Loginova et al. [23], but shifted to lower Ca content in the work of Loginova et al. [23]; similar trends are found by Caviglia et al. [15] but for the three finer grain sizes, which show higher Ca content than expected following the trend of Parma (Figure 2a). From the analyses of smaller grains, it appears that Ca from different plants changes similarly with S, with an apparent deviation from the main trend (Figure 2b). Pb and Zn follow a similar trend in the work of Loginova et al. [23] and in this work, but they follow a different trend in the work of Caviglia et al. [15] (Figure 2c). No correlation between S and Cr is found in any studies. It is likely that these differences in chemical composition may come from the waste input and different burning processes of the WtE plants, despite the fact that a correlation between final composition and waste input may be very difficult to predict with current knowledge.

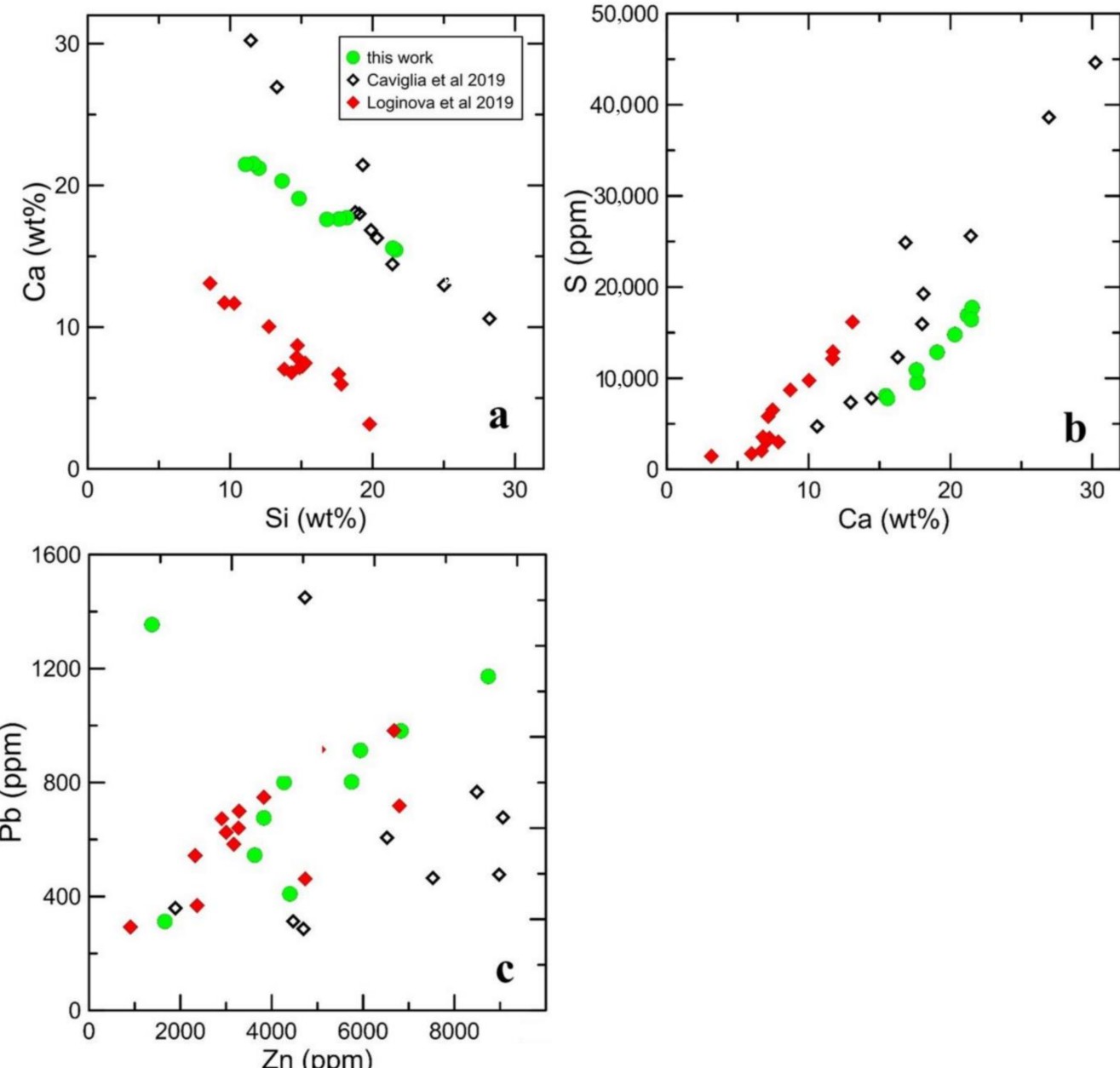

**Figure 2.** Comparison between (**a**) Si vs. Ca, (**b**) S vs. Ca, (**c**) and Pb vs. Zn values for this work (green bullet), data from Loginova et al. [23] (red squares), and data from Caviglia et al. [15] (dark squares).

### 3.3. XRPD and Rietveld Analysis

All of the X-ray powder diffractions show heterogeneous assembly of several crystalline phases, together with significant glass content (Figure 3). To identify the mineralogical phases, in addition to the standard X-ray diffraction pattern made on the different grain-sized portion, selected on the basis of their aesthetic appearance, they are taken from the total sample and discussed. The main goal of this procedure is the characterization of specific phases that are hardly recognizable due to their small quantities inside the sample. Microscopic images of the isolated clasts and their identification with XRD analysis are reported in Table S2. Their composition is very heterogeneous and is represented by small pieces of green, brown, or transparent glass (point 3); white spongy granules of hydroxyapatite of bone residues (point 5); pieces of non-combusted piece of metal (point 4); and gray

aggregates with red or dark areas, which are the typical product of the burning processes. The mineral phases found in these grey clast resemble those noted in the bulk sample and most of the portions after sieving.

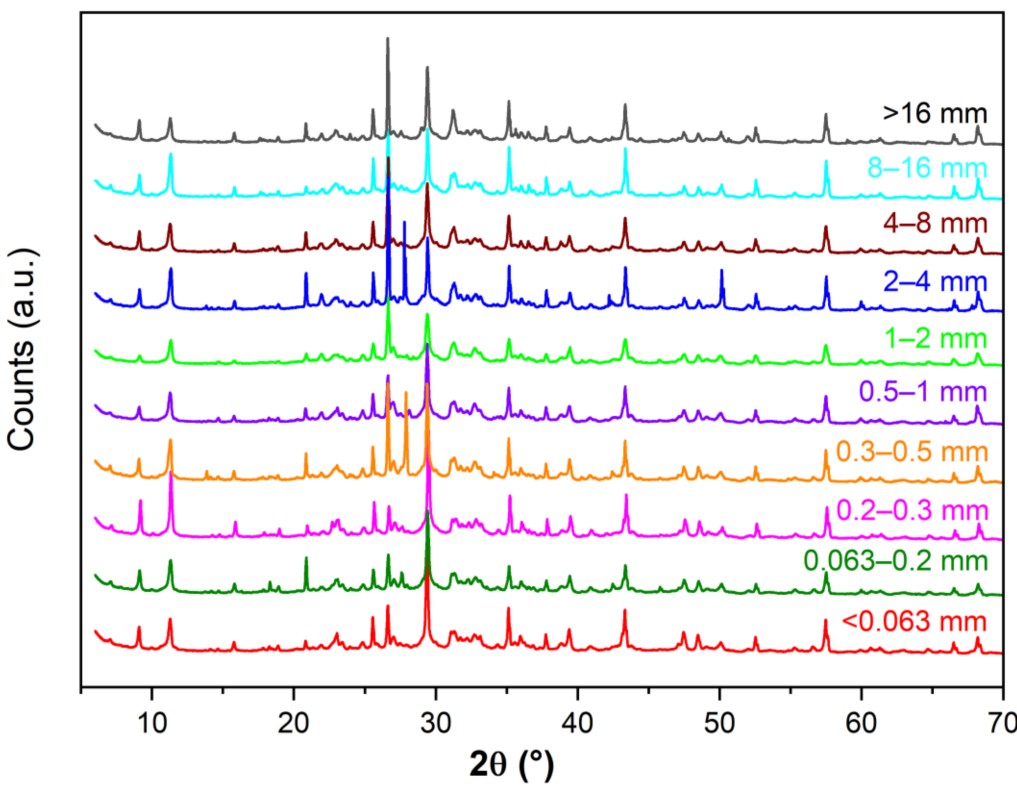

**Figure 3.** XRD patterns made on different grain sizes. Alumina ($Al_2O_3$) standard is used. Major crystalline phases identified are reported. c: calcite, q: quartz, g: gehlenite, e: ettringite, h: hydrocalumite, *: alumina standard, s: strätlingite.

Phase identification was biased by the number of overlapping peaks, which concealed the peaks of the minor phases. Moreover, isomorphic substitutions in plagioclase, gehlenite, phosphates, and sulfates give rise to a shift in peak positions, again hindering correct identification. In Table 2, quantitative Rietveld analysis was conducted on the few phases that could be univocally identified. The results of the Rietveld refinement are given in weight percent normalized to 100%, including the amorphous fraction estimated with the aid of the corundum internal standard.

As shown in Table 2, the amorphous phases are the main constituents in BA. The amorphous phase is higher in the larger grain sized portions, between 75 and 70 wt%, and lower in the smaller ones, between 60 and 65%. The residual glass likely explains the higher amorphous phase content in the larger grains. Crystalline silicates (quartz and melilites) are present in higher percentage in larger grains, whereas carbonates, calcite, and vaterite occur more in smaller sized fractions. As expected, this agrees with the chemical analytical results, but as discussed below, most Ca and Si are present within the amorphous phases. Ettringite is concentrated more in the finer fractions, whereas less abundant hydrocalumite, strätlingite, and iron oxides appear to be unrelated to grain size.

Calcite is the only crystalline phase present with a concentration higher than 10%. Its concentration in lower grain sized portion is likely related to the small size of crystals, which are formed during the carbonation process in air.

Compared to other investigations, a difference in a minor content of feldspar is evident [11,15,23], whose place is partially taken by Ca-Al-Si phases, such as ettringite,

strätlingite, and hydrocalumite. Moreover, sylvite, another commonly found phase, is missing here.

The amorphous phases here are more present than those in the works Bayuseno and Schmahl [11] and Alam et al. [14], who found an amorphous of 33 and 36 wt%, respectively, but they are similar to those identified by Caviglia et al. [15], i.e., about 70%. An intermediate value was found by Wei et al. of about 50% [13].

### 3.4. Bottom Ash Morphology

#### 3.4.1. Optical Observation

Optical observation was conducted on few grains, chosen due to their different color and luster from the portion larger than 4 mm. Similar grains were also analyzed by XRD (Table S2). The grain color varies with the crystalline and amorphous phase content, as revealed by XRD. The few whitish grains str made of hydroxyapatite, likely by the burning of cemetery residuals; green or colorless glass come from residual glass, which did not react during incineration. The rounded reddish-gray grains, which are most abundant, (here and in Chimenos et al. and [5,6] Wei et al. [13]) are mostly made by heterogeneous aggregation of residual refractory material and metals coexisting with crystals and amorphous phases (Figure 4). The amorphous phase acts as a matrix that binds the crystalline and metallic fractions of the original waste and the newly formed minerals. The original waste grains show rounded edges, indicating partial reabsorption (Figure 4a), whereas the newly formed crystals are often sharp needles within the glass, like the wollastonite crystals in Figure 4b. Vesicles and bubbles appear, indicating that some degassing occurs (Figure 4c,g).

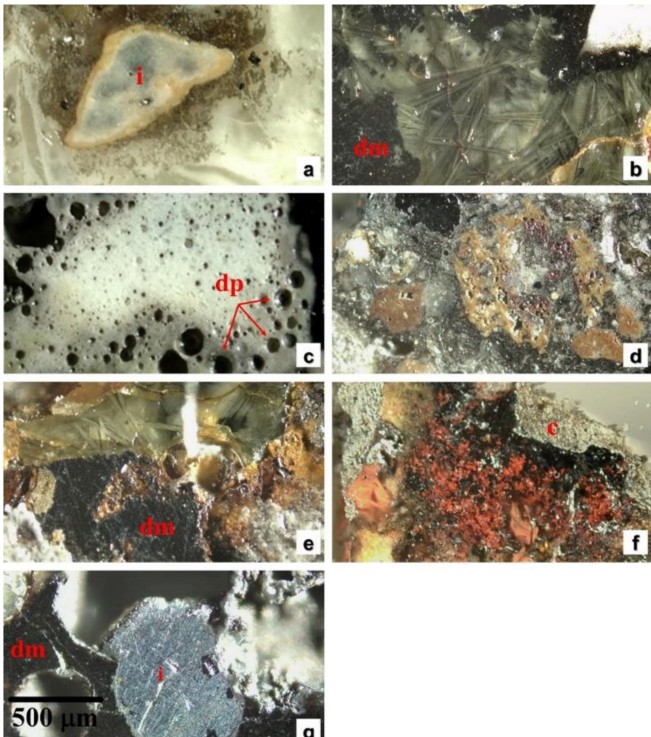

**Figure 4.** Photomicrograph showing the inner part of the clasts cut longitudinally. Each of them appears different from the other in terms of morphology but frequently with part of a dark matrix (dm) (**b**,**e**,**f**,**g**) and, within it, sometimes white (**c**), yellow/red (**e**,**f**,**d**), or transparent (**b**,**e**) areas. Many samples present little vesicles and bubble structures, indicating degassing processing (dp), (**c**) and metallic, alloy, or refractory inclusions (i) (**a**,**g**), sometimes with reaction rims (**a**). External grey crust (**c**) surrounding related to carbonation is shown in figure (**d**).

XRD and subsequent SEM-EDS analysis showed that the grey grains take their color from the coexistence of gehlenite and wollastonite mixed with metal, whereas the red portions take their color from Fe oxides (Table 2 and Table S2). Secondary carbonation is present as an external, fine grey cohesive material that coats the rounded clasts (Figure 4f).

### 3.4.2. SEM-EDS Investigation

SEM backscattered electron images show almost invariable crystals within an amorphous matrix. The prevailing amorphous was investigated in further detail. About 400 EDS point analyses were conducted on different amorphous areas, sampling five polished sections with a size of about 8–10 mm; in each of the five sections, a total of 20 areas were examined. The point analyses on the amorphous are plotted in the ternary diagram $SiO_2$-$CaO$-$Al_2O_3$ in Figure 5, together with the bulk XRF composition for the nine granulometric classes (red crosses).

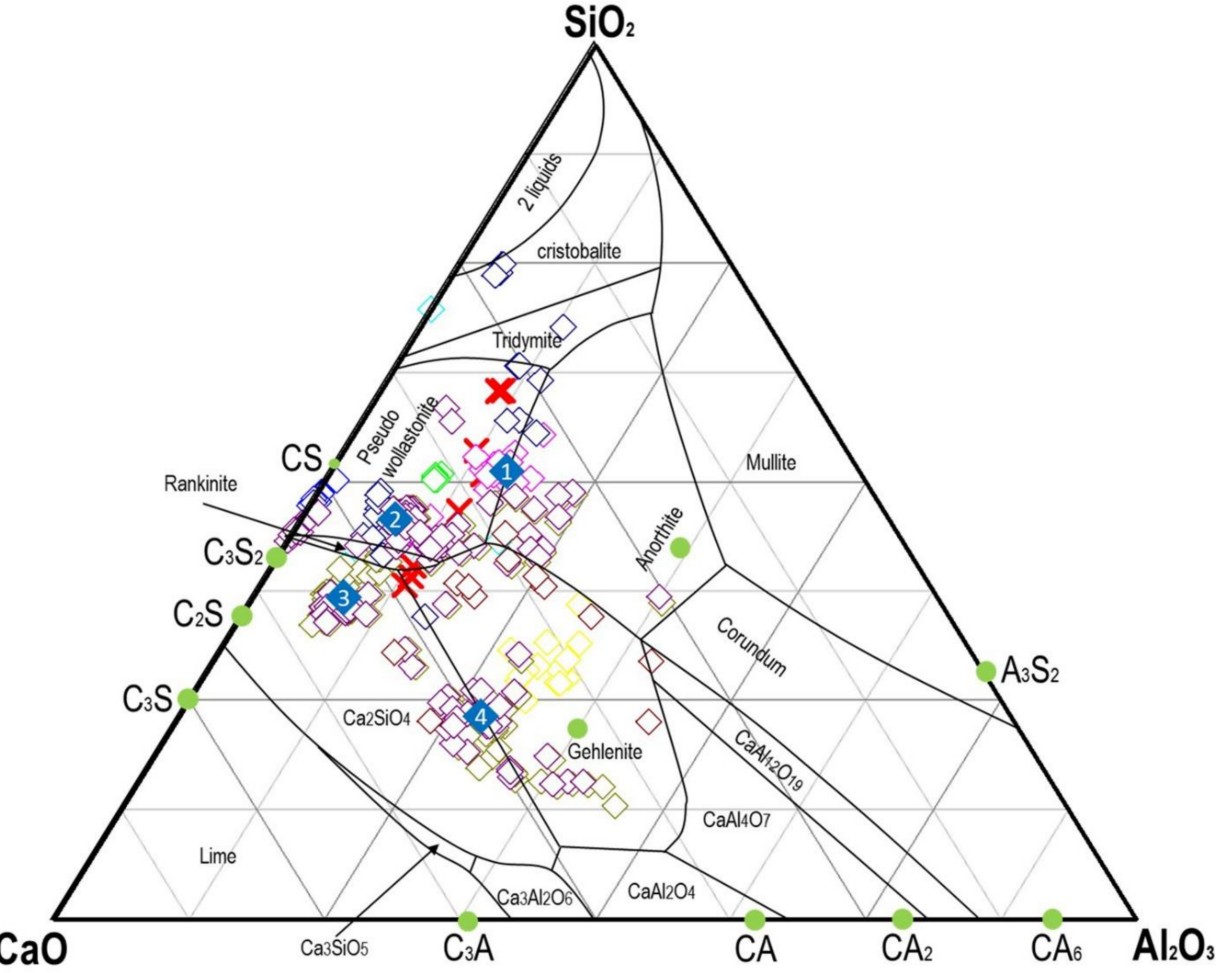

**Figure 5.** Phase diagram of the CaO-SiO2-Al2O3 system originally determined by Rankin and Wright [36] and modified to representing the different regions where a given oxide component crystallizes from the corresponding melt. Specifically, cement notation and stoichiometric formula of some compounds are given for the different phases that can be present in common cement clinkers and ceramics (green points). Empty squares represent point analysis on the amorphous material of different samples (different color); full squares represent the average composition of some cluster points. Red crosses are bulk XRF compositions for the nine grain size classes considered in this work.

The amorphous material is made of an extremely heterogeneous composition, coexisting unmixed at the sub-millimeter scale, even within the same grain. This kind of mingling, well known in igneous petrology, is related to the crystal structure of silica-rich and silica-

poor melts, occurring here at the microscale [37]. The composition of the amorphous within each volume is homogeneous (Figure 4). Optically, this is apparent in the coexistence of reddish and whitish glasses. The chemical composition of the glassy materials is reported in the $SiO_2$-$CaO$-$Al_2O_3$ ternary diagram. Most amorphous compositions fall near or above the cotectic lines with crowding on the ternary minimum; this suggests that the amorphous glass forms from melt at eutectic conditions. About 80% of the analyses falls within the field cornered by pseudowollastonite–gehlenite–larnite. As shown in Figure 5 the average composition of the larger grains is Si-richer than almost all of the analyzed glasses. This likely occurs as XRF analyses also consider pure $SiO_2$ residual glass, which is discarded in the analysis of grains. These results represent the first suggestion of the chemistry of BA and their potential reuse: within grain size separation, two or more chemical compositions can be investigated from the perspective of reuse. In past decades, many studies have focused on the reuse of bottom ash in civil engineering as lightweight aggregates [38], road base materials [39], concretes [40], and sorbents [41].

Many authors have noted the importance of evaluating the long-term environmental impact using MSWI bottom ash added to cementitious materials [42]. From this perspective, grain size and chemical composition characteristic are an important point that could open new possibilities for more suitable reuse. It could be useful to investigate the possible reuse of coarse particle sizes in silica-demanding materials or a smaller grain size if the final product requires significant calcium content.

The glassy matrix is the host of Na, K, and Mg elements, which are not found in crystalline phases reported by XRPD (Table 2) but are present in the XRF analyses between 1 to 5 wt% (in oxide).

Small crystals of silicate and oxides, often with skeletal quenching structures, are found to coexist with the molten matrix. Amongst them, the most common are melilites, minerals that in nature crystallize at high temperatures from alkaline-rich magmas. Natural melilite is a solid solution between akermanite ($Ca_2MgSi_2O_7$) and gehlenite ($Ca_2Al_2SiO_7$). Here, significant Fe is present; thus, the average formula is $Ca_2FeAl_{0.6}Si_{1.4}O_7$. Moreover, some Na and K are substituted for Ca in melilites, which was also previously observed by Bayuseno et al. [11] in soda-melilites. In this work no Cr, Ni, Cu, or Zn was found in melilite, which differs from the observations of Alam et al. (10–15% of the total content) [14].

Among other silicates, quartz, pyroxenes, pyroxenoid, and plagioclase are found. Except quartz, they are hardly found in the XRD pattern, as their major peaks are overlapped by more common phases. Plagioclase is bytownitic, with an average chemical formula of $Na_{0.2}Ca_{0.8}Al_{1.8}Si_{2.2}O_8$, while pyroxene is found with two different compositions, one hedenbergitic ($CaFeSi_2O_6$) and the other augitic $Ca_{0.9}Na_{0.1}(Fe_{0.3}Al_{0.7})Si_2O_6$. Skeletal wollastonite ($CaSiO_3$) and rankinite ($Ca_3Si_2O_7$) are also present (Figure S1a–c).

Among the secondary minerals, the most present are the carbonates, calcite, and vaterite. They appear as incohesive and fine fragments around the clasts (Figure 6e). Other mineral phases ascribable to secondary processes are hydrated silicates, aluminates, and sulfates, such as hydrocalumite [$Ca_8Al_4(OH)_{24}(CO_3)Cl(H_2O)_{9.6}$] and ettringite [$Ca_6Al_2(SO_4)_3(OH)_{12} \times 26H_2O$], as well as gypsum and anhydrite; they are reported by Meima et al. [42] to form in a basic pH environment in air-dried material (pH 10–11). Simple ($A_xO_y$) and complex ($A^{2+}B_2^{3+}O_4$) metallic oxides are another important constituent. Most of them are iron oxides, hematite ($Fe_2O_3$), and magnetite ($Fe_3O_4$). Magnetite, which belongs to the spinel group of minerals, is found both pure and with PTE, as well as with Ca being substituted for Fe. Spinel and simple oxides are found as residual resorbed or newly formed crystals in the glassy matrix (Figure 5c,d and Figure 6a). In the newly formed crystals, $Fe^{2+}$ is often partially replaced by Ca. The Ca–Fe substitution was found by Wei et al. [13], who suggest a high-temperature crystal–melt exchange mechanism. Moreover, substitution of $Fe^{3+}$ and $Ti^{3+}$ in the B site of the spinel-type minerals was observed, as well as the entrance of Cr and Zn (Figure 6a,b).

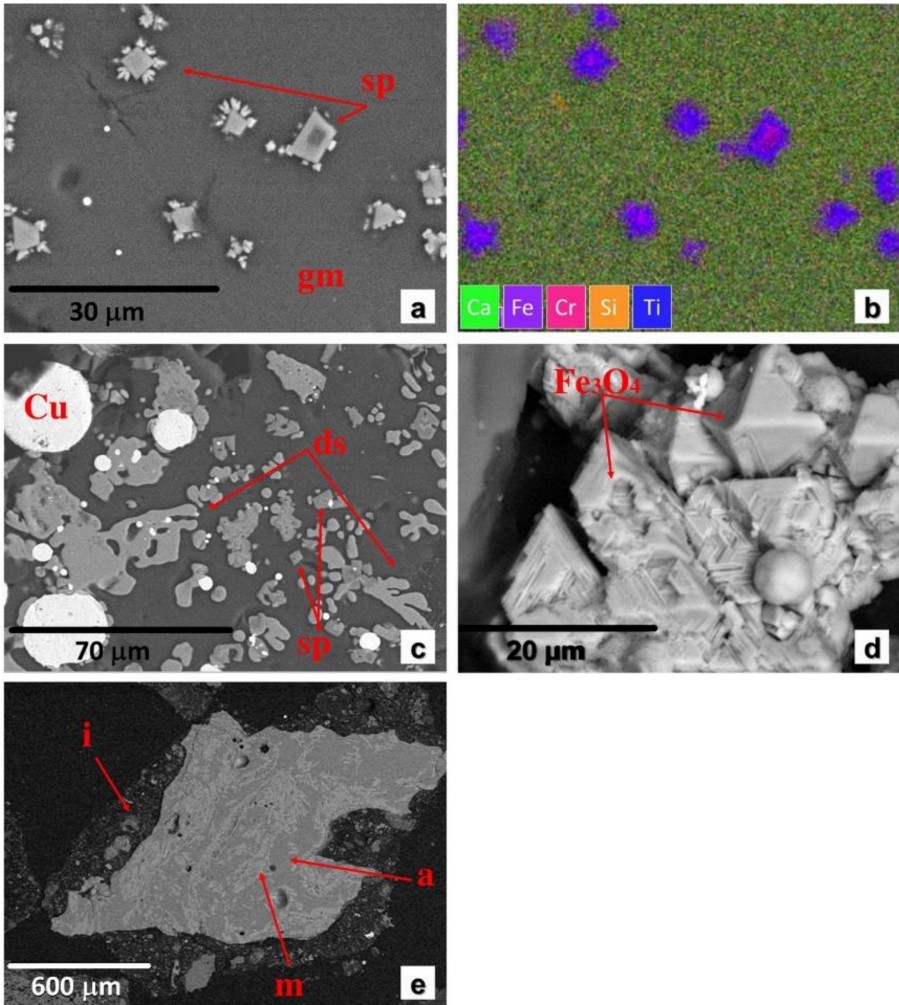

**Figure 6.** SEM images of metallic oxides found in bottom ash samples (**a**) newly formed spinel crystals (sp) in glass matrix (gm) with their (**b**) chemical maps. These crystals are Fe-Ti spinel with an inner core Cr rich; (**c**) dendritic structure (ds) of iron oxides and few little square shape of spinel. White and brightness (at backscattered electron BSE) drops are metallic Cu; (**d**) regular octahedron of magnetite spinel in sample powder; (**e**) large image of a clast: mingling of metallic phases (m) and amorphous inner part (**a**) surrounded by incohesive and fine fragments (if).

Metallic inclusions are present with a size ranging from one micron (or even less) to hundreds of microns (Figure S1d) with spherical or subspherical shapes, as single inclusion or clustered aggregates [43]. Spherical Cu and Fe, or Fe-Ni and Cu-Fe alloys are observed, indicating melting in the incinerator. A Ni-Ti alloy was also found (Figure S1e,f). To note, Cu and Ni where not found other than in the metal inclusions. Cu in this work was most found as metallic copper spheres, sometimes in alloy with iron (Figure 6c).

The secondary minerals are found mainly in the external part of the clast, covering the BA particles and weakly tied to them. These minerals are easily separated from larger grains during transport and sieving due to their breakable nature and low mechanical resistance and accumulate in the smaller fraction. Alteration phases are found despite our samples were sealed in a bottle just a few days after sampling and have not been air-aged like those of Alam et al. [14] and Meima & Coons [44], testifying early ageing process as soon as the bottom ash are quenched in water after incineration.

## 4. Discussion

Combined chemical, SEM-EDS, and XRD investigation with grain size allows one to identify the speciation of major and minor elements. Except Ca and Si, the major elements are poorly correlated with each other. They are present in silicates and non-silicates, although in Parma, in the grain size below 2 mm, Si is positively correlated with Mg, Fe, Al, Na, and K, suggesting that they are present almost completely in silicate glass or crystals.

Fe was found to correlate with other metals, such as Zn, Pb, Cr, and Ni, in the work of Loginova et al. [23]. Such a trend suggests that Fe is present as a metal or oxide, as the same elements are not correlated with Si. In Parma and Caviglia et al. [15], a higher entrance of Fe in silicate glass is probable, as suggested by an Fe–Mg-trend: Fe and Mg are substituted in silicate and oxides, not in metals. We expect that Fe–Mg increases if it is present in silicates or oxides. This is the case in this work and, albeit with poorer correlation, in the work of Caviglia et al. [15], but not in that of Loginova et al. [23].

Ti, commonly given as a major element, shows opposite behavior to Si. In this work, it is present in spinels and metal alloys.

Although Ca is found in silicates, as in melilite, plagioclase, strätlingite, and wollastonite, the major concentration of Ca is in the smaller fraction, and it is related to calcite-vaterite and ettringite, both Si free phases, or spinel-like oxides.

Regarding minor elements, the positive trend with Si and grain size for Zr and Rb indicates that they enter mostly as silicates. More elements are positively related to Ca: S, Cl, Ti, Sn, Zn, Pb, Sr, Ba, Mn, As, and LOI, and in the works of Loginova et al. [23] and Caviglia et al. [15], also Ni and Cu. This suggests that the positive correlation of Ca with so many elements comes from different sources of enrichment in lower sized grains. First, calcite–vaterite phases increase in the smaller grain size, such as elements hosted in the carbonate structure, such as Ba, Zn, or Pb. The same elements are substituted with Ca in sulfates, such as ettringite and gypsum, again seemingly more present in the smaller portion. Part of Cl is likely explained in the LOI: in silicates and glass, Cl is minimally present. We found Ti, Mn, and Ni with Fe and Ca in spinel oxides but not in silicates. Cu is present as pure metal or in alloy with Fe. When present, Cu occurs at high concentrations according to EDS analysis (from 40 to 100 wt.%), with granules usually smaller than 10 μm. Moreover, correlation with Ca in the works of Loginova et al. and Caviglia et al. suggests that Cu may enter as a carbonate [15,23].

However, regarding this general picture, we suggest that for a given element, speciation may occur from different sources. An example is Zn, which was found throughout microprobe analysis in the following forms:

(1)  In solid solution in the spinel oxide with a remarkable heterogeneity between different granules. Zn is substituted with $Fe^{2+}$ between 2000 to 8000 mg/kg.

(2)  As a secondary constituent in glasses of probable cement origin with silicates or aluminates. There is substantial heterogeneity in the chemical composition of the glass, more or less rich in Al or Si, as well as Ca and Fe. Zn is present in quantities at the limit of detection with punctual microprobe analyses, always less than 2000 mg/kg. Given the widespread presence of glass, it may account for a background value, always present at about 1000–2000 mg/kg. It may be the prevailing form in larger grains.

(3)  The presence of Zn in large amounts was observed only in a single sample as a hydroxide, pure or with Ca (up to 74 wt.% in the pure hydroxide and 32 wt.% in that with Ca). As this mineralogical phase needs to be supported by other cases, its presence is not found here (Figure 7).

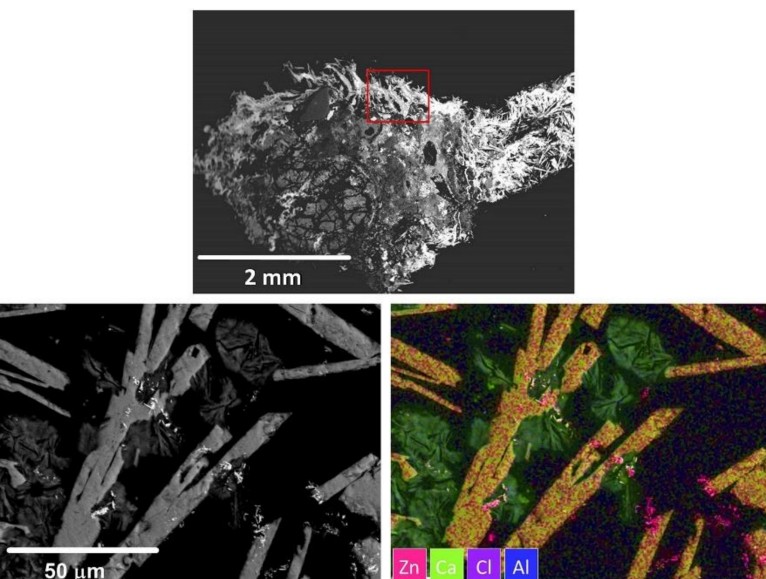

**Figure 7.** Optical and BSE images of a sample area with Zn compound. Chemical map is reported.

## 5. Conclusions

Bottom ashes can be divided into nine different grain sizes (from <0.063 to 16 mm). The size at about 4 mm divides approximately into two portions of similar weight. XRD showed that the amorphous phases are prevailing in any portion at about 60–70% of the weight and more so in the larger one. This is a general feature, even if the amount changes substantially in different incinerators. The mineralogical composition of the crystalline phases shows little differences between the grain size, with higher amounts of quartz in the coarser size (from 3 to 10 wt%) and calcite in the finer one (from 7 to 10 wt%). The presence of ettringite and hydrocalumite suggests that an ageing process may take place in all samples, although they can be considered not exposed to weathering.

We found in minor elements two different trends: a Ca-trend, followed by LOI and a series of minor elements, and a Si-trend, positively related to grain size and lithophile elements, such as Zr and Rb. This can be interpreted in terms of the abundance of crystalline mineral, and of the different amorphous phases present. Due to the prevalence and reactivity of the amorphous phases, further investigation should discuss the different amorphous, their nature, whether they are silicate or not, and their capability to incorporate transition elements.

Furthermore, we demonstrated that bottom ashes, despite the different waste composition, show similar enrichment in elements. The assessment of concentrations and trends of concentration of the elements provide not only an indication of waste composition, local use, and separation efficiency but also information on incinerator metabolism. This, in turn, may suggest procedures to reduce environmental impact.

A few conclusions can thus be drawn:

(a) In bottom ashes, the same crystal phases are observed at any grain size, albeit in different amounts. However, at the electron microscope scale, we observed a wealth of local situations, including residual material made of silicate and metallic inclusions, silicate melts of different composition, droplets of metal phases, and neo-formed minerals, with a variety of quenching structures.

(b) The higher presence of amorphous phases at any grain size suggests that most of the observed compositional changes occur within the amorphous phases.

The presence of heavy metals in the finer fraction, suggested by numerous authors, was observed for only a few elements here. A separation process applied to the finer

fractions to reduce the total heavy metal content would not reduce the overall presence significantly, this fraction being volumetrically almost negligible in terms of output flow.

**Supplementary Materials:** The following are available online at https://www.mdpi.com/2071-105 0/13/4/1911/s1, Figure S1: SEM images of particles catted longitudinally: (a) melilites, (b) pseudowollastonite, and (c) quartz crystals inside an amorphous matrix; (d) fine and heterogeneous materials that cover the BA particles core and a metallic part inside a clast with their original shape; (e) BSE image of a metallic inclusion and their chemical maps (f). Note: the inclusion is made of a strip structure of Ni and Ti, which react partially with the amorphous matrix. Table S1: Linear Pearson's *r* (lower triangle) and Spearman's *p* rank correlation (upper triangle) for evaluating the correlation grade between elements and grain size (log10) for (a) this work and for those of (b) Caviglia et al. [15] and (c) Loginova et al. [24]. Correlation table for <2 mm grain size with major elements also reported (d). Pearson's *r* represents the linear correlation, $r = 0$ no correlation, $r = 1$ completely correlated; Spearman's rank is significant (correlated) if $p < 0.05$. Table S2: XRD analysis: phase identification and semiquantitative estimation in XRD pattern of bulk sample (1) and color-based clast selection: (2) gray, (3) green/transparent, (4) metallic part, (5) white, and (6) red/dark. The semiquantitative contents are represented as xxxx = 70–100 wt%, xxx = 40–70 wt%, xx = 10–40 wt%, x $\leq$ 10 wt%, * = uncertainty presence. Estimated error $\pm$ 10.

**Author Contributions:** Conceptualization, V.F.; Data curation, M.T. and C.D.M.; Investigation, L.M.; Methodology, C.D.M.; Supervision, M.T. and V.F.; Validation, L.M.; Writing–original draft, L.M.; Writing–review & editing, M.T. and V.F.. All authors have read and agreed to the published version of the manuscript.

**Funding:** This research has been financially supported by the program "FIL -Quota Incentivante" of the University of Parma and PRIN 2017 2017L83S77_005 "Mineral reactivity, a key to understand large-scale processes: from rock forming environments to solid waste recovering/lithification". In addition, this work has benefited from the equipment and framework of the COMP-HUB Initiative, funded by the "Departments of Excellence" program of the Italian Ministry for Education, University and Research (MIUR, 2018–2022).

**Institutional Review Board Statement:** Not applicable.

**Informed Consent Statement:** Not applicable.

**Data Availability Statement:** The data that support the findings of this study are available from the corresponding author, LM, upon reasonable request.

**Acknowledgments:** Thanks are due to Barchi Luca and Andrea Comelli from the University of Parma, who provided assistance in thin section mounting and SEM-EDS analysis, and Enrico Dinelli of the University of Bologna for granting access to the XRF laboratory of the BiGeA Department.

**Conflicts of Interest:** The authors declare no conflict of interest.

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
