# Peer review of "Particle Size and Potential Toxic Element Speciation in Municipal Solid Waste Incineration (MSWI) Bottom Ash"

_sustainability, doi:10.3390/su13041911_

Round 1

Reviewer 1 Report

Dear authors,

The manuscript "Particle size and potential toxic elements speciation in MSWI bottom ash" is good applied research. This is an interesting and detailed study about the influence of grain size and potentially toxic elements in the distribution of toxic elements in municipal solid waste incinerators.

This paper deserves to be published but with a major revision in terms of Methodological information. I think that the manuscript would be a valuable contribution to the Sustainability journal.

I recommend a major revision of the manuscript following the comments below.

The XRD discussion of the results needs to be expanded. Also, my expectations were to see some XRD diffractograms, but in the supplementary material (Table S2), only close-up photos and a "semi-quantitative" classes are shown. Also, the XRF subchapter must be improved (see comments within .pdf). Other minor recommendations are mentioned in the manuscript.

Author Response

Dear reviewer,

first of all I would like to thank for the precise revisions and careful reading of our work. In the manuscript all the suggestion you have recommend have been included. You can find them underlined in red in the attached word file.

In particular, you can found:

  • An expanded version of the XRD discussion both for the analyses made on single selected clast and on the grain size portion.
  • A new figure named Figure 3 with all the XRD patterns made on different grain size.
  • The name and the formula of the major mineralogical phases found in XRD analyses in Table 2
  • An improvement of the XRF methodology part 
  • The correction of the minor revision suggested in the PDF.

You can found all the changed part  in red or in yellow (depending if the revision is suggested also by  other reviewers).

Best regards,

Luciana

Reviewer 2 Report

The manuscript presents high quality of structure with a logical coherence; the results are well presented and motivated also providing strong arguments in the discussion part. 

The work takes into account the current studies on the field research area, moreover expanding the knowledge on the bottom ash waste characterization for further reutilization and possible recycling. Although the low/average originality of the work-as also the authors stated in the manuscript, many authors already investigated the mineralogical characterization of MSWI bottom ash with a focus on the potential toxic heavy metals ( also considering leaching tests results before providing possible recycling methodologies)- The authors here have done a complete work by reporting the PTE mineralogy and concentration in relation to the grain size of the MSWI BA, also reviewing similar works and discussing the different outcomes. They clearly report all the methodology they used that can be , hence, easily reproduced also from other authors and they have statistically presented and discussed their results.

Author Response

Dear reviewer,

I would like to thank for the  careful reading of our work and the positive opinion declared. 

Best regards, 

Luciana

Reviewer 3 Report

Manuscript ID: sustainability-1090317

Title: Particle size and potential toxic elements speciation in MSWI bottom ash

Authors: Luciana Mantovani et al.

The article is well-formed. The introduction and methods are described in great detail. The quality of the optic and SEM images is very high.

The data obtained in this work looks reliable. They will expand knowledge of the waste received and give information for further developing of recycling methods. The authors have done a great job. This article corresponds to Sustainability level and should be published after some shortcomings have been corrected.

Title: What is mean MSWI?

Abstract: PTE and MSWI abbreviations must be written in full.

Line 38. What’s wastes can not recycle?

Line 55. Mineral composition of what? Bottom ash?

Line 82. What is mean he WtE of Parma? The plant located in Parma, Italy?

Line 92. bottom ash pile located at the plant or at the landfill? What plant generated this BA? Add location information. How much ash (bottom and fly) are generated per year and total for all time? What material is combustion in this plant?

Table 2. Authors must add XRD patterns of samples with different particle size. Please, write the full chemical formula of all minerals in the table.

Figure 3, 5. Authors can add arrows and phases names (abbreviated). In the figure caption, the Authors can write abbreviations in full.

Figure 4. There are a lot of points on this figure: squares and crosses, all of them in different colours. Leave only those points that belong to your BA. Why only 4 blue squares in the figure, if the figure title writes 9 samples?

Figure 5e, 6a. The scale bar is very difficult to see.

Authors should write a few paragraphs on potential ways to recycle this type of ash. It is low in alumina and high in calcium oxide content. What are the most promising approaches? Use 2018-2020 research when looking for information.

In the conclusions, Authors need to write the values that were obtained in the research: particle size, mineral and chemical composition (highlights) of BA. More numbers need to be added.

Author Response

Dear Reviewer, 

first of all I would like to thank for the precise revisions and careful reading of our work. In the manuscript all the suggestion you have recommend have been included. You can find them underlined in yellow in the attached word file.

Manuscript ID: sustainability-1090317

Title: Particle size and potential toxic elements speciation in MSWI bottom ash

Authors: Luciana Mantovani et al.

The article is well-formed. The introduction and methods are described in great detail. The quality of the optic and SEM images is very high.

The data obtained in this work looks reliable. They will expand knowledge of the waste received and give information for further developing of recycling methods. The authors have done a great job. This article corresponds to Sustainability level and should be published after some shortcomings have been corrected.

Title: What is mean MSWI?

Ok, I explain and add the meaning

Abstract: PTE and MSWI abbreviations must be written in full.

Ok, done

Line 38. What’s wastes can not recycle?

Add and explained in the text

Line 55. Mineral composition of what? Bottom ash?

Yes, corrected in the text

Line 82. What is mean he WtE of Parma? The plant located in Parma, Italy?

Yes, corrected in the text

Line 92. bottom ash pile located at the plant or at the landfill? What plant generated this BA? Add location information. How much ash (bottom and fly) are generated per year and total for all time? What material is combustion in this plant?

More information about the plant and the waste burned are added in the text

Table 2. Authors must add XRD patterns of samples with different particle size. Please, write the full chemical formula of all minerals in the table.

Yes, done

Figure 3, 5. Authors can add arrows and phases names (abbreviated). In the figure caption, the Authors can write abbreviations in full.

New images with arrows and abbreviated phases name are inserted in the manuscript.

Figure 4. There are a lot of points on this figure: squares and crosses, all of them in different colours. Leave only those points that belong to your BA. Why only 4 blue squares in the figure, if the figure title writes 9 samples?

All the empty squares are point analyses made by SEM-EDS on amorphous matrix of some clast with a dimension of about 1 cm of diameter, while red crosses represent XRF ones made on different grain size. The plenty squares point in the ternary diagram represent an average composition of some cluster point measured in the amorphous part with EDS analysis. The caption is slightly changed to better explain the meaning of the figure.

Figure 5e, 6a. The scale bar is very difficult to see.

Ok, I change the colour of the scale bar

Authors should write a few paragraphs on potential ways to recycle this type of ash. It is low in alumina and high in calcium oxide content. What are the most promising approaches? Use 2018-2020 research when looking for information.

Ok, I add your suggestion in the text.

In the conclusions, Authors need to write the values that were obtained in the research: particle size, mineral and chemical composition (highlights) of BA. More numbers need to be added.

Ok, I added some more numbers to better explain.

Round 2

Reviewer 1 Report

Dear authors,

Before I agree with the publication, one more recommendation for figure 3 (XRD patterns): it will be very useful for the readers to follow the diffractograms if you add some symbols/capital letters to the peaks, for the main crystalline phase (quartz, calcite, etc.). If so, please add also a legend to the figure/or add explanations in the figure caption.
Also, please remove hypen from row #248 "plagio-clase" -> plagioclase.

Best regards.

Author Response

Dear reviewer, 

your new suggestions are right! I correct the figure 3 and the caption, 

Best regards, 

Luciana
